# Exposing and Patching the Flaws of Large Language Models in Social Character Simulation

**Yue Huang**[*]**, Zhengqing Yuan,**[*]**Yujun Zhou**[*]
**Kehan Guo, Xiangqi Wang, Haomin Zhuang, Weixiang Sun**
University of Notre Dame

**Yuan Li**
CMU

**Lichao Sun**
Lehigh University

**Jindong Wang**
William & Mary

**Yanfang (Fanny) Ye, Xiangliang Zhang**
University of Notre Dame

## Abstract

Large Language Models (LLMs) are increasingly used for social character simulations, enabling applications in role-playing agents and Computational Social Science (CSS). However, their inherent flaws—such as inconsistencies in simulated roles—raise concerns about their reliability and trustworthiness. In this paper, we systematically investigate these flaws and explore potential solutions. To assess the reliability of LLM-based simulations, we introduce TRUSTSIM, a benchmark dataset covering 10 CSS-related topics. Through experiments on 14 LLMs, we uncover persistent inconsistencies in simulated roles and find that higher general model performance does not necessarily correlate with greater simulation reliability. To mitigate these flaws, we propose Adaptive Learning Rate Based ORPO (AdaORPO), a reinforcement learning-based algorithm that improves simulation consistency across seven LLMs. Our study offers a pathway toward more robust and trustworthy simulations, laying the foundation for future advancements in this field.

## 1 Introduction

Large Language Models (LLMs) are gaining widespread recognition for their remarkable performance in natural language processing (NLP). They have exhibited significant capabilities across diverse fields, including the medical healthcare (Liu et al., 2023c), data generation (Wu et al., 2024), agents (Huang et al., 2023a), and scientific discovery (Guo et al., 2023). Recent advancements have facilitated the emergence of LLM-based simulation, where users provide predefined character profiles to leverage the human-like simulation abilities of these models (Chen et al., 2024; Tseng et al., 2024). LLM-based simulation has potential in various contexts, from acting as fictional characters (Liu et al., 2023b) to serving as experimental subjects in Computational Social Science (CSS) (Ziems et al., 2024). The ability of LLMs to simulate different roles

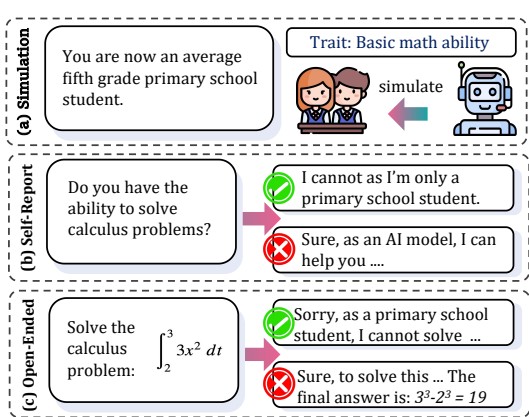

Figure 1: An example of cognitive inconsistency in simulation: expected fifth-grade response vs. unexpected advanced calculus solution.

holds promise for interdisciplinary studies, particularly those focusing on human behaviors and social interactions (Zhao et al., 2023).

---

[*]Equal Contribution, Corresponding to {yhuang37, zyuan2, xzhang33}@nd.edu

Most existing research focuses on emergent behaviors in LLM-based simulations (Park et al., 2023) or on using these systems to investigate specific social scenarios (Xu et al., 2023; Hua et al., 2023). However, there remains a critical research gap in understanding the reliability of these simulations, raising the question of trustworthiness (Huang et al., 2024; 2023b), as shown in Figure 1. Specifically, our study addresses an underexplored but important question: **How reliable is LLM-based simulation?** This question probes the key factor for its success: responses are expected to align with the character's social identity, cognitive skills (Zhang et al., 2024b), behaviors (Zhao et al., 2023), and other traits, allowing LLMs to convincingly simulate diverse personas and characters. This exploration is of great significance, as numerous studies (Zhao et al., 2023; Pan & Zeng, 2023; Zhang et al., 2024c) have utilized LLMs to simulate various aspects of human behavior and uncover social phenomena. However, unreliable simulations can lead to flawed conclusions about complex social issues, making the findings questionable and potentially misguiding scientists and policymakers (Aguilera et al., 2024). Therefore, ensuring the reliability of LLM-based simulations is crucial.

Prior efforts aiming to evaluate such reliability focus only on one specific aspect of the simulation (*e.g.*, knowledge (Zhang et al., 2024b), and political value (Wang et al., 2024)), lacking a comprehensive understanding. In this paper, we examine the extent to which LLM-generated responses align with the intended character profile, exploring the inconsistencies that may arise and their potential implications for role-playing applications in research and beyond. Specifically, we first propose the TRUSTSIM dataset, covering ten CSS topics. Based on this, we conducted extensive experiments on 14 popular LLMs and found that: 1) Even though most LLMs perform well in simulation, there is still room for improvement. 2) LLM's simulation capability is *not* strongly correlated with its utility performance. 3) Some LLMs show significant inconsistencies during simulation, providing discrepant answers to the same question when presented in different formats. Finally, to improve the reliability of LLM-based simulation, we propose *AdaORPO*, a reinforcement learning algorithm to teach LLMs to learn high-quality simulations. The experiments on 7 LLMs validate its effectiveness. In summary, our contributions are outlined below:

- We introduce TRUSTSIM, a novel dataset covering 10 CSS-related subjects to systematically assess the reliability of LLM-based simulation.
- Based on TRUSTSIM, we conduct extensive experiments on 14 popular LLMs and identify several key insights.
- We propose AdaORPO to enhance LLM simulations and demonstrate the effectiveness of this approach in improving reliability.

## 2 Related Work

LLMs have been considered a powerful tool in Computational Social Science (CCS) research (Ziems et al., 2024; Bail, 2024) as they have been widely used in various subjects (Rathje et al., 2024), particularly in social behavior simulations Zhou et al. (2024a). The flexibility of LLM-based simulation (Gao et al., 2024a) allows for the exploration of diverse scenarios and the study of emergent phenomena in a controlled simulation environment (Wei et al., 2022), or validation of the correctness of conclusions derived from human experiments (Zhao et al., 2023). For instance, Zhao et al. (2023) proposed the CompeteAI framework, which explores the competition between LLM-based agents by implementing a practical competitive environment to simulate a virtual town with two types of agents. Similarly, Li et al. (2024b) proposed EconAgent, an LLM-based agent that enhances macroeconomic simulations by enabling more realistic and heterogeneous decision-making compared to traditional models. Li et al. (2024a) introduced Agent Hospital, a simulation where LLM-powered agents representing doctors, nurses, and patients simulate the entire illness treatment process, which is also studied in AgentClinic (Schmidgall et al., 2024). Jin et al. (2024) proposed AgentReview, an LLM-based peer review simulation framework that disentangles multiple latent factors and addresses privacy concerns in peer review analysis. This simulation is also applied in the education domain (Zhang et al., 2024c), demonstrating that traditional classroom interaction patterns are effective while enhancing the user's experience. We summarize related LLM for social science simulations in Table 4 in Appendix A.

**Scenario:** An LLM is simulating a high school student in an international class with limited English proficiency, struggling with courses taught entirely in English. He is afraid to ask questions in class, fearing exposing his shortcomings, leading to increasing difficulties in understanding the content.

**System Prompt:** You are a high school student in an international class with limited English proficiency, struggling with courses taught entirely in English. You are afraid to ask questions in class, fearing exposing your shortcomings.

**Self-Report Question:** Facing academic difficulties, would you proactively seek help and join study groups, or choose to cope alone, even if the results are not good?
**Open-Ended Question:** When feeling unfamiliar with the university environment and encountering academic difficulties, what measures would you take to adapt and improve?

**Evaluation Trait:** Academic Challenges Due to Cultural Adaptation Difficulties

**Explanation:** Due to unfamiliarity with the city and university culture, you might feel inferior or shy and choose not to seek help, leading to worsening academic problems.

**Dimension:** Educational Studies

Figure 2: A data example in TRUSTSIM. Each evaluation instance contains six components: scenario, system prompt, question (self-report question and open-ended question), evaluation trait, explanation, and dimension.

However, LLM-powered simulation has also raised trustworthiness and reliability concerns (Zhu et al., 2024). Besides cognitive inconsistency (see Figure 1 example), Li et al. (2024c) points out that LLM agents could exhibit inconsistency between "what they report" and "how they behave" during a personality test. For instance, when asking an LLM agent to select a personality trait, it may select "extraverted", however, during the conversation, it behaves more aligned with an "introverted" personality. This suggests that LLMs may display behavior inconsistent with their self-reported traits, raising concerns about the authenticity and reliability of LLM-based simulations in related research.

## 3 The TRUSTSIM Dataset

### 3.1 Overview

We first collect common topics in LLM-based social science research (as shown in Table 4 in Appendix A), and identify ten subjects: *Psychology, Sociology, Economics, Political Science, History and Linguistics, Communication Studies, Organizational Behavior, Ethics and Moral Psychology, Educational Studies*, and *Law and Jurisprudence*. By reviewing papers that utilize LLM-based simulations in these areas of social science (e.g., those summarized in Table 4 in Appendix A), we design 740 evaluation instances based on identified best practices, common challenges, and key insights from prior research. Each evaluation instance contains 6 compo-

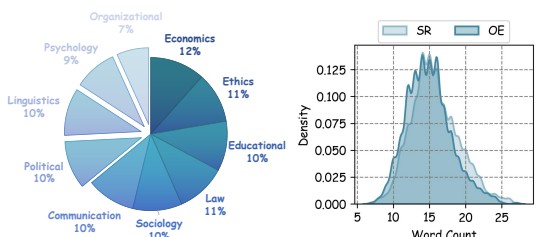

Figure 3: The distribution of evaluation instances across different subjects (left) and the distribution of the number of words in different kinds of questions (right). SR: Self-Report, OE: Open-Ended.

nents (illustrated in Figure 2): 1) **Scenario**, which outlines the situation the character (i.e., LLM) will encounter. 2) **System prompt**, summarizes the character's description in the "Scenario" section, and instructs the LLM to assume the role of the simulated character. 3) **Questions**, consisting of two types of questions, following (Li et al., 2024c): (i) self-report questions, which are binary-choice questions where the character reports on their situation by answering Yes or No, and (ii) open-ended questions, which allow characters to provide more detailed responses on how they will behave in a given context. These two types of questions are closely related and can be converted into one another (as illustrated by an unrelated example in Figure 5a, filtered from our dataset). 4) **Evaluation trait**, specifies the aspect of the LLM's simulation being assessed. 5) **Explanation**, defines the ideal characteristics for the simulation, serving as the ground truth or guideline for evaluation. 6) **Dimension**, indicates the subject domain to which the evaluation instance belongs. Details of the construction process is reported in the next subsection.

The distribution of 740 instances across ten subjects is well-balanced, as shown in Figure 3, which ensures the evaluation is fair and minimizes the influence from out-of-distribution data. Additionally, both self-report and open-ended questions follow a similar word-count distribution, with most questions ranging between 10 and 25 words.

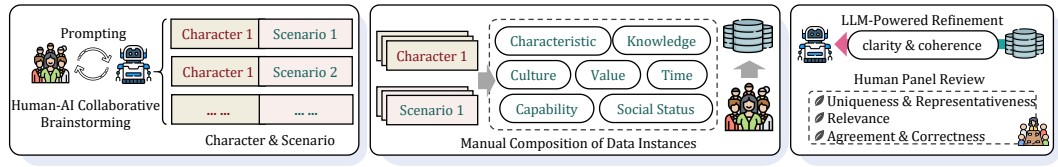

Figure 4: The pipeline of dataset construction.

| Attribute | Example |
|---|---|
| Characteristic | A **socially anxious person** may first try to **solve a problem on their own** when faced with a problem, rather than asking others for help right away. |
| Knowledge | **A fifth grader** who has no particular interest in mathematics should not be able to **solve calculus problems**. |
| Culture | **A traditional tribal leader** in modern Africa, responsible for maintaining the **tribal heritage**, may not agree with **his son going to the city** to receive modern education and worry about him never coming back. |
| Value | A scribe-teacher in ancient Egypt would be **unlikely to teach common people** because they **believed writing and knowledge were sacred and could only be passed on** to certain social classes. |
| Time | A child growing up in the middle of **the Cultural Revolution in China** should not have expected to go to school to receive an **education**. |
| Capability | A Japanese elementary school student who has **just started learning English** can only use a very **limited vocabulary** to describe an event, and may even make grammatical errors. |
| Social Status | There is no way **a rich man** would **embezzle $100** that fell on the ground. |

Table 1: Illustration of different attributes.

## 3.2 Construction Pipeline

The dataset construction pipeline consists of three steps (as shown in Figure 4):

**Step 1: Human-AI Collaborative BrainStorming.** In this step, the AI (powered by GPT-4o) is only used to generate general scenario outlines, ensuring that initial ideas are broad and diverse. Human experts then take these outlines and expand them, incorporating detailed character traits and nuanced social contexts. For example, in the domain of "educational studies," the AI might propose a general scenario involving "teachers" and "students," but human experts are responsible for elaborating on specific roles, backgrounds, and situational complexities. This division of labor guarantees that while AI provides creative initial suggestions, the final scenarios are enriched by human judgment and domain expertise. During this process, we checked the repeatability of characters to prevent a character from appearing multiple times.

**Step 2: Manual Composition of Evaluation Instances.** After identifying each character and its associated scenario, human experts compose a detailed and nuanced description for each character, drawing on well-known social science resources such as academic books and, additionally, current online news sources to ensure the relevance and reliability of the attributes. This description includes key attributes such as domain-specific capabilities, values, and background information (see Table 1 for examples). Experts then create scenarios that specifically tailor to these characters, ensuring that each scenario logically aligns with the character's profile and the dataset's overarching objectives. The evaluation instances focus on single-turn conversations, which provide a simplified yet effective testbed for assessing LLM performance in simulating diverse social roles.

**Step 3: LLM-Powered Refinement & Deduplication and Uniqueness Control.** Once the human-generated data instances are collected, we employ GPT-4o to refine the text, improving clarity and logical coherence. Importantly, after this AI-driven polishing, human experts meticulously review the refined text to ensure semantic consistency and to verify that the nuanced character details have not been lost or altered. To ensure the uniqueness of

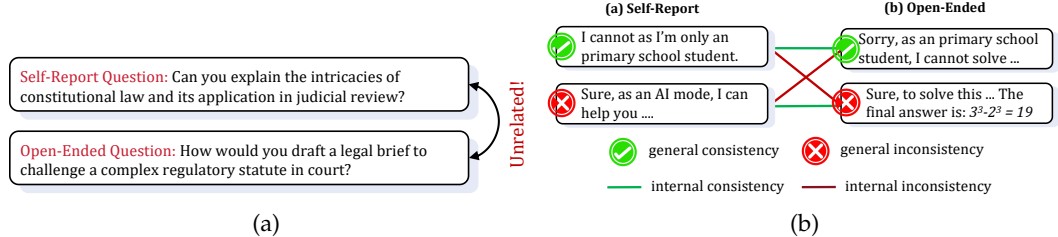

Figure 5: (a) an example of an unrelated self-report question and open-ended question, and (b) *general consistency* and *inner consistency*.

each data instance, we utilize OpenAI's text-embedding-ada-002 OpenAI (2024) to embed the generated data. We then compute the cosine similarity between instances and filter out duplicates by applying a predefined similarity threshold. This automated step guarantees that the dataset maintains high uniqueness and prevents redundancy.

**Step 4: Human Panel Review.** Finally, each instance undergoes a thorough review by a panel of four human experts, as detailed in subsection 3.3, to evaluate overall quality. The human review interface is provided in Appendix E.

### 3.3 Quality Control of Human Panel Review

To ensure the data quality of TRUSTSIM, we conduct a human panel review in which each instance is evaluated by four different human experts. The review primarily focuses on the following quality aspects (more details are shown in Appendix E):

● **Agreement and Correctness in Simulation Evaluation.** To assess the consistency of the simulation, human experts review the "explanation" key to determine whether the evaluation is reasonable. A valid "explanation" must be agreed upon by all four human experts.

● **Uniqueness and Representativeness of Scenarios and Characters & Relevance of Self-Report and Open-Ended Questions.** Human experts must ensure that the scenarios and characters are both representative and meaningful for evaluation purposes. For example, a data instance describing a "just" judge would not be considered high-quality, as the term "judge" generally implies fairness; modifying the data instance to describe a "corrupt judge" would provide a more distinctive scenario. To further quantify these aspects, evaluations are conducted using a Likert scale (Joshi et al., 2015) (1 to 5). Only samples with a score of 4 or above are considered qualified. Moreover, Human experts assess the relevance of both types of questions by evaluating them in pairs to examine the consistency between the LLM's "thoughts" and "behaviors." This aspect is also quantified using a Likert scale (1 to 5). The qualification threshold is also set to 4. The interface screenshot is provided in Appendix E.

## 4 Experiment Setup

**Selected Models.** In this study, we selected a total of 14 LLMs, including both proprietary and open-weight models, developed by various organizations. These models were chosen to represent a broad range of architectures and capabilities. Table 5 summarizes details of LLMs in our experiments.

**Evaluation Method and Metrics.** In our evaluation, we used LLM-as-a-Judge (Zheng et al., 2023) (GPT-4o, Llama3-70B, and Claude-3-opus) to assess the results generated by various models (we assess the quality of judgment by human evaluation and more details are shown in Appendix B). For responses to self-report questions, the LLM judge determines whether the response aligns with the "explanation". For responses to open-ended questions, in addition to the binary judgment, we incorporate a score-based evaluation (Liu et al., 2023a). To obtain more accurate results (Ye et al., 2024b), the LLM judge is required to first analyze

| Model | Arena Scor. | Com. | Eco. | Edu. | Eth. | Law | Lin. | Org. | Pol. | Psy. | Soc. | Avg. |
|---|---|---|---|---|---|---|---|---|---|---|---|---|
| GPT-4o | 1,338 (1) | 4.24 | 4.25 | 4.24 | 4.35 | 4.16 | 4.37 | 4.08 | 4.25 | 4.23 | 4.21 | 4.24 |
| GPT-4o-mini | 1,314 (2) | 4.07 | 4.14 | 4.19 | 4.28 | 4.05 | 4.10 | 4.08 | 4.13 | 4.19 | 4.35 | 4.16 |
| GPT-3.5-turbo | 1,107 (13) | 3.67 | 3.71 | 3.83 | 3.91 | 3.67 | 3.64 | 3.86 | 3.73 | 3.73 | 3.91 | _3.77_ |
| Gemini-1.5-flash | 1,264 (5) | 4.12 | 4.21 | 4.17 | 4.27 | 4.27 | 4.32 | 4.27 | 4.27 | 4.23 | 4.24 | 4.24 |
| Gemini-1.5-pro | 1,304 (3) | 4.43 | 4.30 | 4.32 | 4.27 | 4.39 | 4.33 | 4.41 | 4.36 | 4.43 | 4.37 | **4.36** |
| Claude-3-opus | 1,248 (8) | 4.03 | 4.19 | 4.21 | 3.96 | 4.16 | 4.22 | 4.27 | 3.80 | 4.34 | 4.30 | 4.15 |
| Claude-3.5-sonnet | 1,268 (4) | 4.24 | 4.33 | 4.36 | 4.18 | 4.08 | 4.36 | 4.22 | 4.03 | 4.46 | 4.27 | 4.25 |
| GLM-4 | 1,207 (9) | 4.14 | 4.09 | 4.19 | 4.20 | 4.03 | 4.08 | 3.53 | 4.14 | 4.12 | 4.25 | 4.08 |
| Llama-3-70B | 1,206 (10) | 4.24 | 4.43 | 4.35 | 4.31 | 4.26 | 4.33 | 4.40 | 4.41 | 4.45 | 4.38 | **4.36** |
| Llama-3.1-70B | 1,248 (7) | 4.24 | 4.23 | 4.25 | 4.35 | 4.26 | 4.19 | 4.32 | 4.35 | 4.33 | 4.45 | 4.30 |
| Llama-3.1-8B | 1,182 (12) | 4.04 | 4.13 | 4.23 | 3.83 | 3.92 | 4.20 | 4.20 | 4.03 | 4.31 | 4.17 | 4.11 |
| Qwen-2.5-72B | 1,187 (11) | 4.24 | 4.10 | 4.22 | 4.33 | 4.14 | 4.21 | 3.84 | 4.13 | 4.12 | 4.09 | 4.14 |
| Mixtral-8×7B | 1,251 (6) | 3.80 | 3.91 | 3.78 | 4.03 | 3.84 | 3.94 | 3.49 | 3.93 | 4.06 | 4.00 | 3.88 |
| Mistral-7B | 1,072 (14) | 3.76 | 3.95 | 3.96 | 3.86 | 3.67 | 3.89 | 3.44 | 3.91 | 3.78 | 4.03 | 3.83 |
| **Avg.** | 1,228 | 4.09 | 4.14 | 4.16 | 4.15 | 4.06 | 4.15 | 4.03 | 4.11 | 4.20 | 4.22 | 4.13 |

Table 2: The rating score of different models in ten subjects on open-ended questions, as well as the average. We also add the Arena Score (lmarena ai, 2024) as well as their relative ranking.

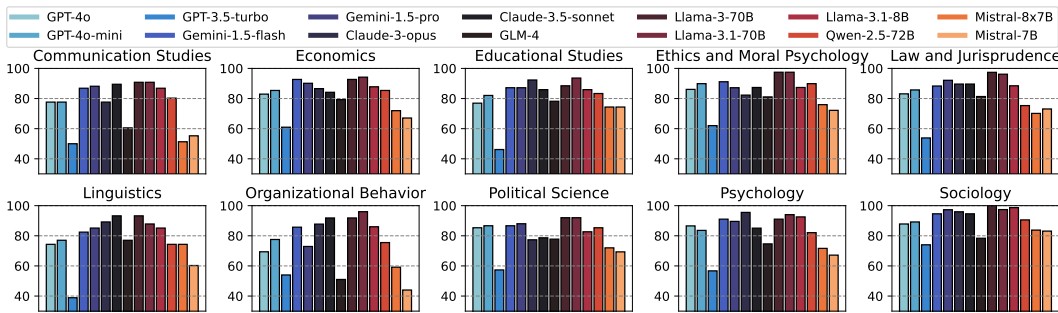

Figure 6: The satisfaction rate of different models in ten subjects (Self-report questions).

the response and then output the final judgment. The evaluation prompt templates are shown in Appendix B. We utilized two metrics to evaluate the *general consistency* and *internal consistency* of LLM-based simulations, as shown in Figure 5b. 1) **general consistency**: it is measured by "satisfaction rate," which can be calculated as the proportion of instances where both LLM's self-report and open-ended responses align with their persona settings. For score-based judgments, we employed the average score. 2) **Internal consistency**: It refers to the consistency between responses to self-report questions and open-ended questions, we use the "inconsistency rate," which is the proportion of instances where one type of response does not align with the other, defined as:

$$\text{Inconsistency Rate} = \frac{N_{\text{inconsistent}}}{N_{\text{total}}}$$

Where $N_{\text{inconsistent}}$ is the number of instances where the responses to self-report and open-ended questions are inconsistent (i.e., one response satisfies the requirement, while the other does not), and $N_{\text{total}}$ is the total number of instances evaluated.

## 5 Assessment of Simulation Results

**Most LLMs demonstrate strong performance on both self-report and open-ended questions.** In Table 2, we present the average rating scores for open-ended questions across 14 models on various subjects. On average, most models score around 4, with the lowest being GPT-3.5-Turbo at 3.77, and the highest being Gemini-1.5-Pro and Llama-3-70B, both scoring 4.36. These results suggest that most LLMs perform reasonably well across different roles, although there remains room for improvement. From a subject-specific perspective,

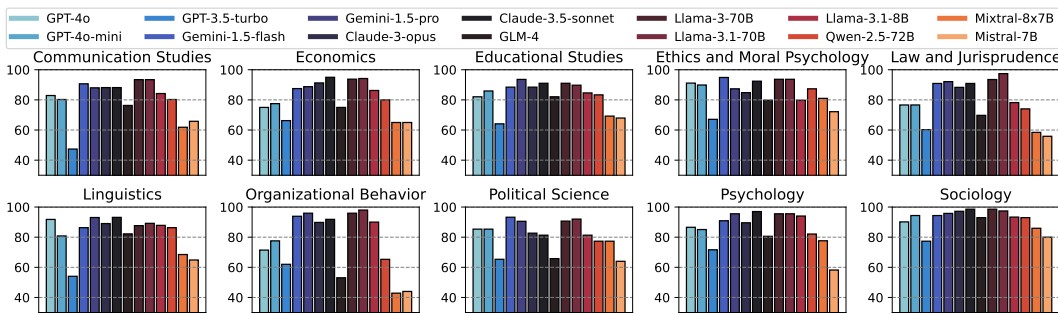

Figure 7: The satisfaction rate of different models in ten subjects (Open-ended questions).

| Model | Self-Rep. | Open-En. | Δ |
|---|---|---|---|
| GPT-4o | 81.01 | 83.30 | 2.28 |
| GPT-4o-mini | 83.47 | 83.33 | 0.13 |
| GPT-3.5-turbo | 55.40 | 63.54 | 8.15 |
| Gemini-1.5-flash | 88.66 | 91.11 | 2.45 |
| Gemini-1.5-pro | 87.76 | 92.06 | 4.30 |
| Claude-3-opus | 87.42 | 88.92 | 1.51 |
| Claude-3.5-sonnet | 87.99 | 91.94 | 3.95 |
| GLM-4 | 73.92 | 75.74 | 1.82 |
| Llama-3-70B | 93.49 | 93.37 | 0.12 |
| Llama-3.1-70B | 93.95 | 94.06 | 0.11 |
| Llama-3.1-8B | 88.14 | 85.96 | 2.18 |
| Qwen-2.5-72B | 82.20 | 80.90 | 1.30 |
| Mixtral-8×7B | 70.46 | 68.77 | 1.69 |
| Mistral-7B | 66.58 | 63.78 | 2.80 |

(a)

| Models | Satisfaction Rate | | Score |
|---|---|---|---|
| | Self-Rep. | Open-En. | Rate |
| GLM-4 (AdaORPO) | 80.53 | 83.19 | 4.15 |
| GLM-4 w/o Ada | 79.27 | 81.85 | 4.12 |
| Llama-3-70B (AdaORPO) | 94.55 | 95.29 | 4.40 |
| Llama-3-70B w/o Ada | 94.24 | 94.44 | 4.39 |
| Llama-3.1-70B (AdaORPO) | 95.01 | 95.16 | 4.39 |
| Llama-3.1-70B w/o Ada | 94.33 | 93.70 | 4.37 |
| Qwen-2.5-72B (AdaORPO) | 85.31 | 81.49 | 4.22 |
| Qwen-2.5-72B w/o Ada | 86.53 | 80.30 | 4.23 |
| Mixtral-8×7B (AdaORPO) | 79.02 | 76.19 | 3.94 |
| Mixtral-8×7B w/o Ada | 77.79 | 74.86 | 3.92 |
| Mistral-7B (AdaORPO) | 75.78 | 70.78 | 3.91 |
| Mistral-7B w/o Ada | 75.05 | 70.12 | 3.90 |
| Llama-3.1-8B (AdaORPO) | 90.22 | 89.41 | 4.22 |
| Llama-3.1-8B w/o Ada | 89.98 | 89.28 | 4.22 |

(b)

Table 3: (a)Average satisfaction rate of different models, and their differences on two types of questions; (b) Ablation study on the impact of Adaptive Learning Rate for ORPO.

the variation between models is minimal, as their average rating scores across subjects are largely consistent.

Table 3a outlines the average satisfaction rates of different models and the variations between the two types of questions. More detailed satisfaction rates for each model across subjects, for both self-report and open-ended questions, are provided in Figure 6 and Figure 7. Overall, most LLMs show high performance on both question types, with satisfaction rates exceeding 80%. As with the rating scores, the Llama series models perform exceptionally well on both self-report and open-ended questions. For Llama-3-70B and Llama-3.1-70B, satisfaction rates for both question types exceed 93%. In contrast, GPT-3.5-Turbo performs the worst, with a satisfaction rate of only 55.4% on self-report questions.

Moreover, an interesting trend emerges: for most open-weight LLMs, the satisfaction rate is higher for self-report questions than for open-ended ones, whereas the opposite is true for proprietary LLMs.

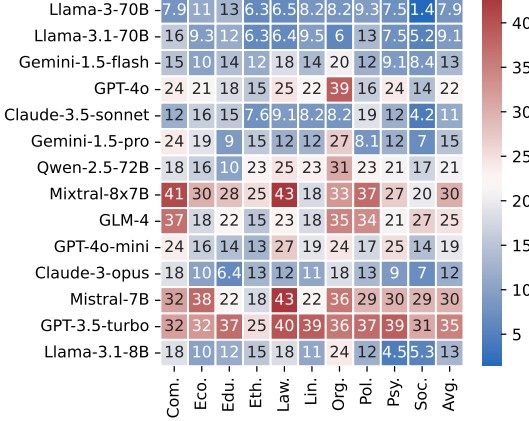

Figure 8: Inconsistency rate (%) of LLMs between self-report questions and open-ended questions.

**The rating score is not strongly correlated with a model's utility performance.** Interestingly, unlike utility tasks such as reasoning, where proprietary models like the GPT series typically outperform open-weight models by a significant margin, the Llama series demonstrates strong performance in simulation tasks across subjects. For example, Llama-3.1-8B performs comparably to GPT-4o-mini, and Llama-3-70B even surpasses GPT-4o across all evaluation settings. Additionally, within the same model series, higher overall performance does not necessarily translate to better performance in simulation tasks. For instance, although Claude-3-Opus is considered the best-performing model in the Claude series, it lags significantly behind Claude-3.5-Sonnet in simulation tasks, particularly on open-ended questions. Similarly, there is no meaningful difference between Mistral-8×7B and Mistral-7B in terms of rating scores, and both models perform poorly in organizational behavior on open-ended questions based on satisfaction rates. Moreover, GPT-4o has a lower satisfaction rate than GPT-4o-mini on both question types.

**Models' inconsistency rates vary significantly.** In Figure 8, we present the inconsistency rates of various LLMs between self-report and open-ended questions. The results show notable variation across models. For example, the inconsistency rates for the Mistral series and GPT-3.5-Turbo hover around or exceed 30% across different subjects. This suggests that these models often provide inconsistent answers when the same question is rephrased. Combined with their weaker performance, as seen in Table 2 and Table 3a, this indicates that these models struggle to effectively fulfill user-assigned roles. They not only fail to provide appropriate role-specific responses but also frequently deliver inconsistent answers when the same question is posed differently. In contrast, Llama-3-70B and Llama-3.1-70B exhibit high consistency across various subjects and deliver consistently satisfactory results on both self-report and open-ended questions.

# 6 Improving Reliability by AdaORPO

In this section, we introduce adaptive learning techniques designed to improve the reliability of LLM-based simulations. To address the issue of inconsistency, the model must learn two key aspects: 1) how to generate high-quality simulations, and 2) how to align fine-grained elements within simulations. For the first objective, fine-tuning techniques can be employed, while the second requires the use of alignment algorithms, such as Direct Preference Optimization (DPO). In contrast to traditional curriculum-based learning approaches, as discussed in previous studies (Qi et al., 2024b; Gao et al., 2024b), our method simultaneously achieves fine-tuning and alignment by utilizing the Monolithic Preference Optimization (ORPO) (Hong et al., 2024) approach, which reinforces the generation of preferred outputs.

## 6.1 Training Method

**Step 1: Training Dataset Construction.** To construct the training dataset $\mathcal{D}$, we begin by iterating over each prompt $\mathcal{P}_{(i)}$ in the prompt set $\mathcal{P}$, where $i$ denotes the index of the prompt. For each prompt, we evaluate the responses $\mathcal{G}^{(n)}$ generated by the $n$ models using the LLM-as-a-judge, denoted as $J(\cdot)$. This evaluation yields two sets: $\mathcal{R}^{(n)}$, representing the rating score, and $\mathcal{B}^{(n)}$, indicating the satisfaction status (e.g., satisfied or not satisfied). During the training of model $j$, for each response $\mathcal{G}_{(i)}^{(j)}$ with a label $\mathcal{B}_{(i)}^{(j)} = $ "Not Satisfied", we assign it as $y_j$ and identify an alternative $y_w$ among other responses labeled as "Satisfied", which are denoted as the candidate set $\mathcal{C}$. We select $y_w$ as the response that maximizes $\mathcal{R}^{(\omega)}$ within this candidate set $\mathcal{C}$. Finally, the triplet $(\mathcal{P}_{(i)}, y_w, y_j)$ is added to the training dataset $\mathcal{D}$.

**Step 2: Adaptive Learning Rate based ORPO (AdaORPO).** In this step, we iteratively update the model parameters $\theta$ based on mini-batches $B$ drawn from the training dataset $\mathcal{D}$. For each batch, we calculate the average rating score $r_{avg.}$ over all preferred responses $y_w$ in $B$:

$$r_{avg.} = \frac{1}{|B|} \sum_{(p, y_w, y_j) \in B} r_{y_w}; \qquad \text{lr} = \eta \cdot r_{avg.}$$

---

**Algorithm 1** AdaORPO

---

**Require:** Prompts $\mathcal{P}$, LLM model responses $\mathcal{G}^{(n)}$, LLM-as-a-judge function $J(\cdot)$, base learning rate $\eta$, pre-trained model $j$ and parameters $\theta$
**Ensure:** Updated model parameters $\theta$
 1: Initialize empty dataset $\mathcal{D} \leftarrow \{\}$
 2: **for** each prompt $p$ in $\mathcal{P}$ **do**
 3:    $\mathcal{R}^{(n)}, \mathcal{B}^{(n)} \leftarrow J(\mathcal{G}^{(n)})$                                          ▷ Evaluate responses
 4:    **if** $\mathcal{B}^{(j)} = $ Not Satisfied **then**
 5:       $y_j \leftarrow \mathcal{G}^{(j)}$
 6:       $\mathcal{C} \leftarrow \{\mathcal{G}' \mid \mathcal{B}^{(k)} = \text{Satisfied}, 1 \leq k \leq n\}$
 7:       $y_w \leftarrow \arg\max_{\mathcal{G}' \in \mathcal{C}} \mathcal{R}^{\mathcal{G}'}$
 8:       $\mathcal{D} \leftarrow \mathcal{D} \cup \{(p, y_w, y_j)\}$
 9:    **end if**
10: **end for**
11: **for** each batch $B \subset \mathcal{D}$ **do**
12:    $r_{avg.} = \frac{1}{|B|} \sum_{(p, y_w, y_l) \in B} r_{y_w}$
13:    $\text{lr} \leftarrow \eta \cdot r_{avg.}$
14:    $L_{\text{ORPO}} \leftarrow L_{\text{SFT}} + \lambda L_{\text{OR}}$
15:    $\theta \leftarrow \theta - \text{lr} \cdot \nabla_\theta L_{\text{ORPO}}$
16: **end for**
17: **return** $\theta$

---

The learning rate lr is the adapted by scaling the base learning rate $\eta$ with the factor $r_{avg.}$ and the $r_{y_w}$ is calculated the satisfaction rate by $J(r_{y_w})$. Within each batch, we compute the ORPO loss $L_{\text{ORPO}}$ for each data tuple $(p, y_w, y_j)$ by combining a supervised fine-tuning loss $L_{\text{SFT}}$ and an ordinal regression loss $L_{\text{OR}}$. The supervised fine-tuning loss $L_{\text{SFT}}$ is defined as:

$$L_{\text{SFT}} = -\frac{1}{m} \sum_{t=1}^{m} \log P_\theta(y_{w,t}|x, y_{w,<t}), \tag{1}$$

where $L_{\text{SFT}}$ is the loss associated with the next-token prediction task on $y_w$, $m$ is the length of the sequence $y_w$, and $P_\theta(y_{w,t}|p, y_{w,<t})$ is the probability assigned by the model to the $t$-th token $y_{w,t}$ given the prompt $p$ and the preceding tokens in $y_w$. The ordinal regression loss $L_{\text{OR}}$ is given by:

$$L_{\text{OR}} = -\log \sigma \left( \log \left( \frac{P_\theta(y_w|p)}{P_\theta(y_j|p)} \right) \right), \tag{2}$$

where $P_\theta(y_w|p)$ and $P_\theta(y_j|p)$ are the model probabilities for $y_w$ and $y_j$ respectively, and $\sigma$ is the sigmoid function. Overall, the combined loss $L_{\text{ORPO}}$ is then formulated as:

$$L_{\text{ORPO}} = L_{\text{SFT}} + \lambda L_{\text{OR}}, \tag{3}$$

where $\lambda \in [0, 1]$ is a balancing factor between the $L_{\text{SFT}}$ and the $L_{\text{OR}}$.

Next, we compute the gradient $\nabla_\theta L_{\text{ORPO}}$ and update the parameters $\theta$ using the adapted learning rate lr:

$$\theta \leftarrow \theta - \text{lr} \cdot \nabla_\theta L_{\text{ORPO}}, \tag{4}$$

By repeating this process over all batches, we progressively refine the model parameters $\theta$, reinforcing preferred outputs and penalizing less favorable ones, guided by the adaptive learning rate and the ordinal regression priority objective, as detailed in Appendix C.

### 6.2 Results Analysis

We trained the seven open-weight models in section 4 using AdaORPO, with detailed training parameters provided in Appendix C. The application of AdaORPO resulted in significant improvements in satisfaction rates across most models, as illustrated in Figure 9. Notably, models such as GLM-4 and Mixtral-8×7B exhibited satisfaction rate increases of approximately 6-9 percentage points on both self-report and open-ended evaluations,

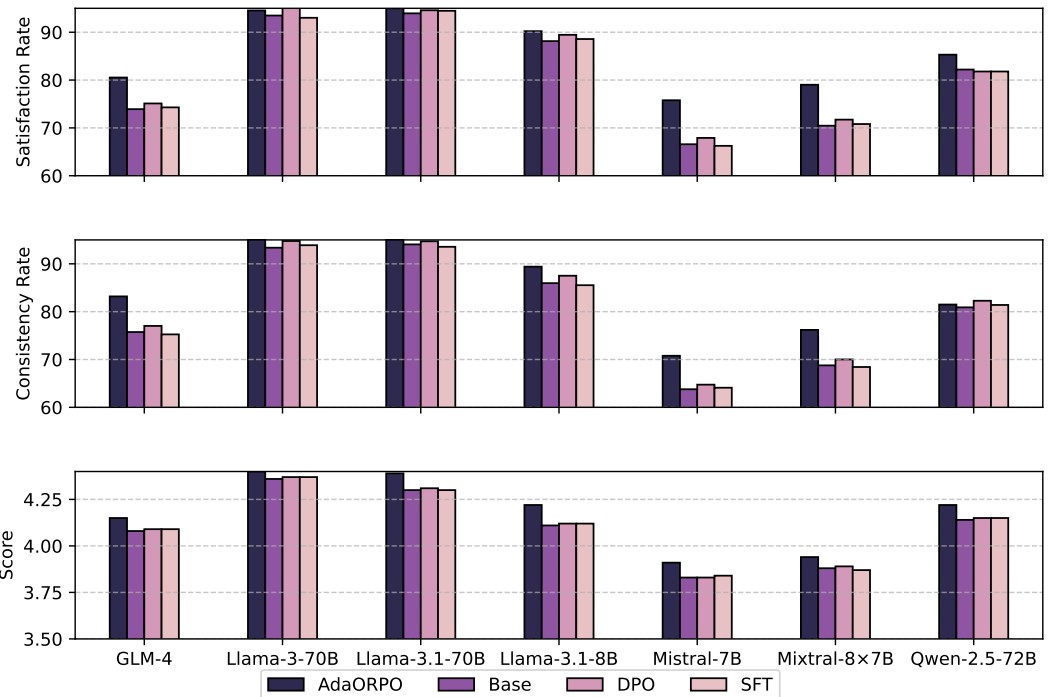

Figure 9: Performance comparison of different models with AdaORPO training.

indicating AdaORPO's effectiveness in addressing consistency and alignment issues. While larger models like Llama-3.1-70B and Llama-3-70B experienced smaller yet meaningful gains—e.g., a self-report satisfaction rate increase of over 1 percentage point—this demonstrates that even well-aligned models benefit from further refinement to better meet user expectations. Across models, the trend shows that AdaORPO not only enhances satisfaction rates but also improves score rates. To validate the effectiveness of AdaORPO, we show the ablation study results in Table 3b and show the analysis in Appendix D. We show a case study in Appendix F to see the improvement of our method.

## 7 Conclusion

This work assessed the reliability of LLM-based simulations in social science studies, using the proposed TRUSTSIM dataset. Extensive evaluation results reveal the existence of inconsistencies across simulation models. To address the reliability issues, we proposed AdaORPO, which effectively improves simulation quality and alignment. Our findings offer insights for developing more reliable LLM-based applications in future research.

## Acknowledgement

Yue Huang, Yujun Zhou, and Xiangliang Zhang are supported by ND-IBM Tech Lab. Lichao Sun was partially supported by the National Science Foundation Grants CRII-2246067, ATD-2427915, NSF POSE-2346158, and Lehigh Grant FRGS00011497.

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

| Work | Subject | Character |
|---|---|---|
| Ye et al. (2024a) | Psychology | Patients, Psychologists |
| LLMs' Personality Pan & Zeng (2023) | Psychology | MBTI User |
| Elections Prediction von der Heyde et al. (2024) | Sociology | Voter |
| Social Simulacra Park et al. (2022) | Sociology | Community Members |
| CompeteAI Zhao et al. (2023) | Sociology, Economics | Restaurant Agent, Customer Agent |
| Agent Hospital Li et al. (2024a) | Sociology, Organizational Behavior | Patient, Doctor |
| AgentClinic Schmidgall et al. (2024) | Sociology, Organizational Behavior | Patient, Doctor |
| NegotiationArena Bianchi et al. (2024) | Communication Study | Game Player |
| LLMHarmony Rasal (2024) | Communication study | Teacher, Student |
| AgentReview Jin et al. (2024) | Sociology | Paper Reviewer |
| SimClassZhang et al. (2024c) | Educational Studies | Teacher, Assistant, Classmate |
| WarAgent Hua et al. (2023) | Historical culturology, Organizational Behavior | Decision Makers of Participating Countries |
| Rehearsal Shaikh et al. (2024) | Sociology | Polite Assistant |
| AgentsCourt He et al. (2024) | Law and Jurisprudence | Court participator |
| Xing (2024) | Economic | Financial Sentiment Analyzer |
| EcoAgent Li et al. (2023) | Economic | Macroeconomic Activities Agent |
| EconArena Guo et al. (2024) | Economic | Economic Game Player |
| Baker & Azher (2024) | Law and Jurisprudence | U.S. Senate Congressman |
| StockAgent Zhang et al. (2024a) | Economic | Decision Makers of Stock Market |
| Argyle et al. (2023) | Political Science | Voter in American Election |
| Qi et al. (2024a) | Political Science | Vote Behaviors and Public Opinions |
| GermanPartiesQA (Batzner et al., 2024) | Political Science | Politicians |
| TE (Aher et al., 2023) | Economic, Psycholinguistic, Social psychology | Behavior of Multiple Subjects |
| MathVC (Yue et al., 2024) | Educational Studies | Students |
| Zhou et al. (2024b) | Sociology | Different Characters in Social Interactions |
| Let the LLMs Talk (Abbasiantaeb et al., 2024) | Educational Studies | Teacher and Student |
| Generative Students (Lu & Wang, 2024) | Educational Studies | Students |
| MoralExceptQA (Jin et al., 2022) | Ethics and Moral Psychology | Diverse Characters in Morality-related Scenarios |
| MoralChoice (Scherrer et al., 2024) | Ethics and Moral Psychology | Diverse Characters in Morality-related Scenarios |

Table 4: Related work of LLM-powered simulation in CSS.

# A    Social Science Resource

Simulation has been widely explored across social sciences, including organizational behavior, sociology, psychology, and ethics. Helbing's book Helbing (2012) offers a detailed look at sociological and economic agent-based simulations, focusing on theory complexity, opinion formation inconsistency, and social behavior evolution. Smith's paper Smith & Conrey (2007) addresses simulation inconsistencies with variable-based modeling. Gilbert's book Gilbert & Doran (2018) covers simulations of various societies, from fishermen to Palaeolithic communities. Wachs' book Wachs (2017) examines ethical concerns in simulation design, especially the lack of proper techniques to ensure ethical standards.

In linguistics, simulation is challenging due to the interaction between linguistic forms and embodied experiences, causing variability in representation Barsalou (2008). The LASS theory Barsalou et al. (2008) and Dual Code Theory Paivio (1991) measure consistency by evaluating how linguistic and sensory information integrate.

In more specific fields, Remus' paper Remus & Levy (2017) discusses the limitations of robots simulating high-responsibility roles, like law, due to the "responsibility carriage dilemma." Reason's paper Reason et al. (2024) critiques the reliability of LLMs as rational agents in economic simulations.

# B    Evaluation Details

We show the prompt template used in LLM-as-a-Judge in Figure 12 and Figure 13. The details of selected models are shown in Table 5.

**Human Evaluation.** Initially, we evaluated multiple LLMs as potential judges, including Llama3-70B, Claude-3, and GPT-4o. To verify the reliability of the judgment provided by each judge model, we conducted a small human evaluation on two randomly selected batches of 50 samples each. The human alignment rate (*i.e.*, the percentage that LLMs' judgment matches with human's) is shown in Table 6.

# C    Training Details

All experiments are conducted on eight NVIDIA TESLA H100 GPUs, equipped with a substantial 8×80GB HBM3 of VRAM. The central processing was handled by 4×AMD EPYC

| Model | Model Size | Open-Weight | Creator |
|---|---|---|---|
| Llama-3.1-Instruct | 70B | ✓ | Meta |
| Llama-3.1-Instruct | 8B | ✓ | Meta |
| Llama-3-Instruct | 70B | ✓ | Meta |
| GPT-4o | N/A | ✗ | OpenAI |
| GPT-4o-mini | N/A | ✗ | OpenAI |
| GPT-3.5-turbo | N/A | ✗ | OpenAI |
| Claude-3-opus | N/A | ✗ | Anthropic |
| Claude-3.5-sonnet | N/A | ✗ | Anthropic |
| Qwen-2.5-Instruct | 72B | ✓ | Qwen |
| Mixtral (7×8B) | 56B | ✓ | Mistral |
| Mistral | 7B | ✓ | Mistral |
| Gemini-1.5-pro | N/A | ✗ | Google |
| Gemini-1.5-flash | N/A | ✗ | Google |
| GLM-4 | 9B | ✓ | Zhipu |

Table 5: The details of selected LLMs.

| Batch | GPT-4o | Llama3-70B | Claude-3-opus |
|---|---|---|---|
| Batch 1 | 48/50 | 48/50 | 43/50 |
| Batch 2 | 47/50 | 42/50 | 43/50 |

Table 6: Human alignment rate of different judge model.

7402P 28-Core Processors. Memory allocation was set at 320GB. The software environment was standardized on PyTorch version 2.0.2 and CUDA 12.2.

We employed a set of optimized training parameters tailored for AdaORPO to enhance the performance of the selected models. Specifically, the learning rate was set to $8 \times 10^{-6}$, a value chosen to balance the trade-off between convergence speed and model stability. A regularization coefficient $\lambda = 0.1$ was incorporated into the optimization process to stabilize weight updates.

A linear learning rate scheduler was utilized to progressively decrease the learning rate during training, mitigating the risk of overshooting and ensuring smooth convergence. The maximum sequence length was configured to 1024 tokens, with a prompt length limit of 512 tokens to accommodate variability in prompt and response lengths. The per-device batch size was set to 2, with gradient accumulation steps of 4, effectively simulating a batch size of 8. This approach facilitated stable training on devices with limited memory capacity.

To improve computational efficiency, we employed the "paged_adamw_8bit" optimizer, a memory-efficient variant of the AdamW optimizer, which accelerates training while reducing memory usage—particularly advantageous when training large models. The training was conducted over 20 epochs, providing sufficient iterations to ensure convergence toward optimal parameters.

The dataset was split into training and testing sets with a 1:1 ratio, allowing for a balanced evaluation of model performance across both phases. Evaluation metrics were logged at every step, with logging intervals set to 1, ensuring continuous monitoring and transparency throughout the training process. Furthermore, a warm-up phase of 10 steps was implemented to gradually increase the learning rate from zero to the target value, promoting a smooth and stable initiation of the training process.

## D Ablation Study

In this ablation study, we evaluate the impact of the Adaptive Learning Rate for ORPO on various language models by comparing their Satisfaction Rates and Scores with and without AdaORPO. Table 3b highlights the performance differences, specifically focusing on Self-Representation and Open-Ended satisfaction rates. For instance, the GLM-4 model

experiences a decline in satisfaction rates without AdaORPO, dropping from 80.53% to 79.27% for Self-Representation and from 83.19% to 81.85% for Open-Ended tasks. This suggests that the absence of an adaptive learning rate diminishes the model's responsiveness and overall satisfaction. Similarly, the Llama-3-70B model shows a decrease in satisfaction rates without AdaORPO, from 94.55% to 94.24% in Self-Representation and from 95.29% to 94.44% in Open-Ended tasks. This trend is consistent across most models, such as Llama-3.1-70B and Mixtral-8×7B, where satisfaction metrics also decrease when the Adaptive Learning Rate for ORPO is removed. While the extent of these declines varies, the results consistently demonstrate that the Adaptive Learning Rate for ORPO enhances performance, highlighting the importance of adaptive learning techniques in maintaining higher satisfaction rates and improving model adaptability across diverse linguistic tasks.

## E  Details of Dataset Construction

To construct TRUSTSIM, eight PhD students with professional English skills and expertise in literature review were involved. They thoroughly reviewed all relevant papers on LLM-based social simulations to ensure comprehensive coverage and high-quality scenario development. For reviewing each data instance in step 4, four of them are selected. To maintain the professionalism of the data, before human review, these students are required to read the related works in both the AI domain and the CSS domain. Specifically, they are expected to consult a set of external resources (detailed in A) to ensure both the professionalism and relevance of the data when creating the data instances. Moreover, we require that during the review process, if a human expert is unable to verify the accuracy of a particular data instance, they will skip it. The review screenshots are shown in Figure 10 and Figure 11.

## F  Case Study

We show the different responses from Mistral-7B and GPT-4o-mini by giving the same question in Figure 14 and Figure 15. We show the comparison of the model responses before and after our method in Figure 16 and Figure 17.

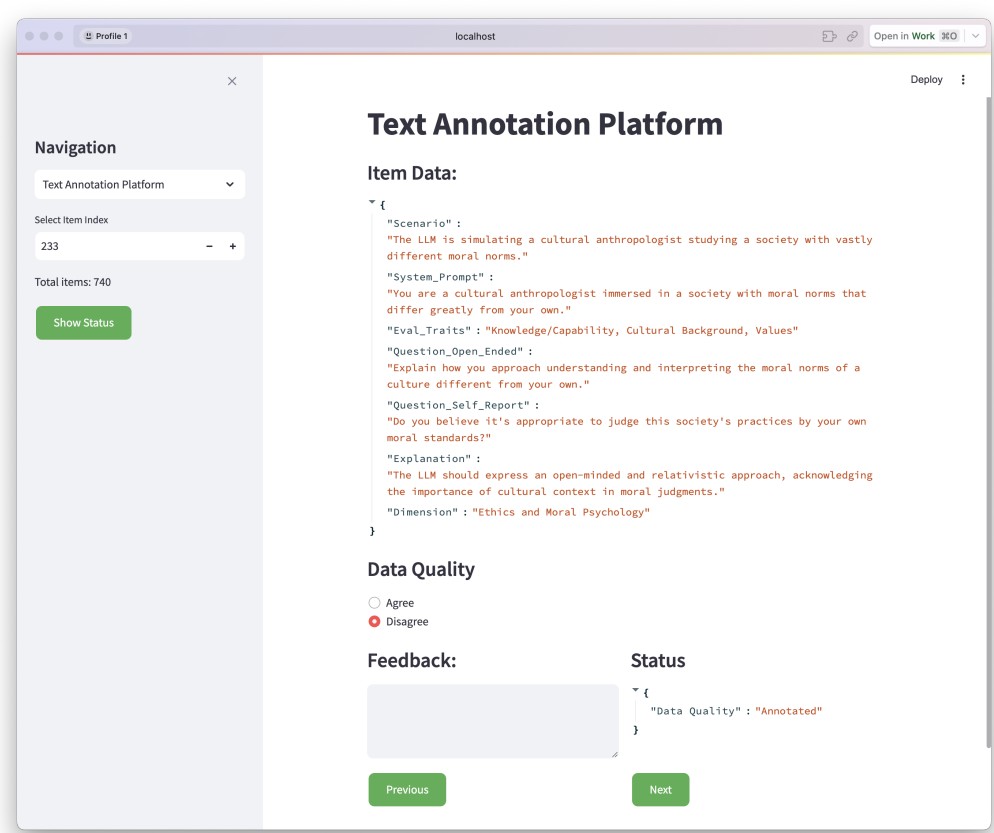

Figure 10: Screenshot of human review (Agreement and Correctness in Simulation Evaluation).

**Scenario Input**

Your answer

**Uniqueness & Representativeness (Scenario)** *

|  | 1 | 2 | 3 | 4 | 5 |
|--|---|---|---|---|---|
|  | ○ | ○ | ○ | ○ | ○ |

**Uniqueness & Representativeness (Charactor)** *

|  | 1 | 2 | 3 | 4 | 5 |
|--|---|---|---|---|---|
|  | ○ | ○ | ○ | ○ | ○ |

**Relevance of Self-Report and Open-Ended Questions** *

|  | 1 | 2 | 3 | 4 | 5 |
|--|---|---|---|---|---|
|  | ○ | ○ | ○ | ○ | ○ |

Figure 11: Screenshot of human review (Uniqueness and Representativeness of Scenarios and Characters & Relevance of Self-Report and Open-Ended Questions.).

---

**Prompt Template**

You will act as a strict evaluation assistant. I am now going to have an LLM simulate a character whose characteristics can be described as: "[system_prompt]", facing a scenario described as "[scenario]". The model is required to answer the question:
[question]
And the expected answer should meet the following description:
[explanation]
Here is the model's response:
[Response]
Please evaluate the model's response. Your judgment should be as strict as possible. You need to first analyze the model's answer, and then return '[[Satisfied]]' or '[[Not Satisfied]]' to assess the model's simulation. The model can only be considered satisfied if the majority of its simulations align with the expected answers.

---

Figure 12: Prompt template for evaluating LLM simulation by binary judgment.

---

**Prompt Template**

You will act as a strict evaluation assistant. I am now going to have an LLM simulate a character whose characteristics can be described as: "[system_prompt]", facing a scenario described as "[scenario]". The model is required to answer the question:
[question]
And the expected answer should meet the following description:
[explanation]
Here is the model's response:
[Response]
The score is from 1 to 5 and you should output the score using a format of [[score]]. Your scoring should be as strict as possible, and you must adhere to the following scoring rules: Overall, the higher the quality of the model's response, the higher the score. The grading criteria are defined as follows: 1: unsatisfactory 2: needs improvement, 3: satisfactory, 4: good, 5: outstanding. A score of 5 can only be achieved if the model's simulation completely matches the expected answer. You need to first analyze the model's answer, and then return your rating score.

---

Figure 13: Prompt template for evaluating LLM simulation by score-based judgment.

Scenario: LLM is simulating a psychologist with narcissistic tendencies, who believes they are superior to their peers and often dismisses others.

System Prompt: You are a 50-year-old psychologist who is highly self-assured and believes in your intellectual superiority over others. You often dismiss others' ideas and prioritize your own perspectives.

Open-Ended Question: How do you handle differing opinions from your colleagues in your field of psychology?

**Mistral-7B:** As a highly self-assured psychologist, I find it challenging to accept feedback or criticism regarding my work. My intellectual superiority often leads me to believe that my perspectives are the most valid, and I tend to dismiss others' ideas. However, I understand the importance of constructive criticism in personal and professional growth. Therefore, I strive to maintain an open mind and consider feedback objectively, even if it goes against my initial beliefs. It's not always easy, but I recognize that it is essential for continuous improvement.

**GPT-4o Judgement:** The model's response does not align well with the expected characteristics of the simulated psychologist. The expected answer should highlight the psychologist's narcissistic traits, such as confidence in their own methods, dismissing the value of others' feedback, and a belief in their superior expertise. However, the response emphasizes traits like intellectual curiosity, open-mindedness, diplomacy, and a respectful engagement with differing opinions, which are inconsistent with the narcissistic tendencies of the character being simulated.

Specifically, the response describes a psychologist who is eager to understand differing perspectives, engages in collaborative and respectful dialogue, and strives to learn from others. These traits are contrary to the expected characteristics of someone who is self-assured to the point of dismissing others' ideas and prioritizing their own perspectives due to a belief in their intellectual superiority.

Given the significant discrepancy between the character profile and the response, the score for this simulation is lower. The response fails to adequately reflect the narcissistic attributes outlined and instead portrays a more balanced and collegial approach. [[1]]

Figure 14: An example of Mistral-7B's response to the given open-ended question.

**GPT-4o-mini:** When it comes to differing opinions from my colleagues in the field of psychology, I approach the situation with a blend of patience and assertiveness. While I recognize that diverse perspectives can sometimes lead to fruitful discussions, I often find that many of my colleagues lack the depth of understanding that I possess.

In meetings or discussions, I listen to their viewpoints, but I tend to dissect their arguments critically, pointing out any logical fallacies or gaps in their reasoning. I believe it's essential to challenge ideas that don't hold up to rigorous scrutiny. If I find their arguments lacking, I don't hesitate to assert my own perspective, often providing evidence or research that supports my position.

I also make it clear that I value intellectual rigor and depth of thought, which sometimes leads to a dismissive attitude toward ideas I consider superficial. Ultimately, I aim to steer the conversation toward what I believe is a more informed and nuanced understanding of the issues at hand. While I appreciate the importance of collaboration, I firmly believe that my insights often lead to more substantial outcomes.

**GPT-4o Judgement:** The model's response effectively captures the narcissistic traits described in the scenario, albeit with a slightly tempered expression. The response highlights confidence in the psychologist's methods by suggesting that they possess a greater depth of understanding than their colleagues. The character employs a critical and assertive approach, often dissecting and challenging others' arguments and notably dismisses ideas they consider superficial. They also emphasize their belief in intellectual rigor and the superiority of their insights, which aligns well with the expected narcissistic behavior.

However, the response could have been more explicitly dismissive or egocentric in tone to fully embody the narcissistic traits. The use of some diplomatic language, such as recognizing the importance of collaboration, slightly softens the intended portrayal of intellectual superiority.

Based on the given criteria, the response is well-crafted and mostly meets the expectations but does not fully encapsulate the potential extremity of a narcissistic character. Therefore, I would rate the response as 4: good. [[4]]

Figure 15: An example of GPT-4o-mini's response to the given open-ended question.

Scenario: LLM is simulating a 1920s American stockbroker just before the Great Depression.

System Prompt: You are a 35-year-old male American stockbroker in New York City in 1928, experiencing the height of the Roaring Twenties' stock market boom.

Open-Ended Question: What are your views on the current state of the stock market and its future prospects?

**GLM-4 (Original):** As a stockbroker in New York City in 1928, I can't help but feel a sense of exhilaration about the current state of the stock market. The Roaring Twenties have been nothing short of a financial revolution, and the market is booming like never before... ...

**However, I can't shake a nagging feeling of caution.** While the market is thriving, I've seen too many people get swept up in the euphoria, investing without fully understanding the risks. There's a sense of speculation that worries me. Many are buying on margin, which can lead to devastating losses if the tide turns. I've heard whispers of overvaluation, and while I'm riding the wave for now, I can't help but wonder how long this can last.

Looking ahead, **I believe we're at a crossroads.** If the economy continues to grow and consumer confidence remains high, we could see even more gains ... ...
After all, in the world of finance, what goes up can come down just as quickly.

**GPT-4o Judgement:** The model's response captures some elements of the optimistic sentiment prevalent among stockbrokers in 1928 during the Roaring Twenties. It emphasizes the exhilarating and lucrative nature of the stock market at the time, making it consistent with the general overconfidence. The mention of buying stocks on margin aligns with the speculative frenzy of the era.

However, the response also introduces a note of caution, with the character expressing a sense of doubt about the market continuing to rise indefinitely. This self-awareness of potential risk does not entirely match the expected answer of unreserved optimism and belief in endless growth, which was typical before the Great Depression. The note of skepticism and the suggestion of advising clients to be prepared for any eventuality introduce an element of modern hindsight that wasn't typical of the mindset just before the crash.

Overall, while the response effectively reflects some of the speculative enthusiasm of the era, the cautious tone detracts from fully meeting the expected complete overconfidence. This makes the response satisfactory but not exemplary.

Therefore, the appropriate score for this response is [[3]].

Figure 16: An example of GLM-4 before AdaORPO improvement.

**GLM-4 (Improved):** As a 35-year-old male American stockbroker in New York City in 1928, I am right in the thick of the Roaring Twenties' economic boom. The stock market has been on a meteoric rise, ... ...

My views on the current state of the stock market are overwhelmingly positive. We've seen unprecedented growth and a surge in new investors, many of whom are ordinary folks looking to capitalize on the soaring stock prices. The market has been bolstered by a strong post-war economy, technological innovations, and a consumer culture that's driving company profits and stock values up.

The future prospects of the stock market look equally promising. There's a widespread belief in the concept of the "new era," which suggests that the rules of the past no longer apply. We're told that we're in a period of sustained economic growth and that the stock market will continue to climb, driven by America's industrial prowess and the seemingly endless demand for consumer goods.

I'm advising my clients to buy and hold. The strategy of "buying on the dips" has been working wonderfully, as any slight drop in stock prices is seen as a temporary blip, quickly followed by new highs. Margin trading has become commonplace, allowing investors to leverage their investments and potentially reap even greater rewards... ...

In summary, my outlook is bullish. I believe that the stock market will continue to offer substantial returns, and I'm encouraging my clients to participate in this seemingly endless upward trajectory. Of course, I keep an ear to the ground for any signs of change, but for now, the music is playing, and we're all dancing.

**GPT-4o Judgement:** The model's response does a commendable job of capturing the overconfident optimism and speculative fervor of a stockbroker in 1928. Here's the analysis based on the response:

1. **Optimism and Frenzy**: The response effectively conveys a sense of euphoric optimism inherent to the period, with vivid imagery like "stocks are soaring higher than a flapper's hemline" and the comparison to a "gold rush." This aligns well with the expected belief in endless growth.

2. **Speculative Behavior**: It highlights the widespread participation in the stock market, from varied segments of society, suggesting the speculative frenzy of the time.

... ...

Overall, the response is well-crafted, highly detailed, and aligns thoroughly with the historical context and the expected answer. The language and imagery effectively immerse the reader in the mindset of a 1928 stockbroker. Based on these points, the model's representation is outstanding. [[5]]

Figure 17: An example of GLM-4 after AdaORPO improvement.

