# OpenReview forum: "Exposing and Patching the Flaws of Large Language Models in Social Character Simulation"
_colmweb.org/COLM/2025/Conference — COLM 2025_

### Official Review · Reviewer_Lamc · 2025-05-02

**Rating:** 7
**Confidence:** 4
**Ethics Flag:** 1

**Summary:**

The paper introduces TRUSTSIM, a benchmark dataset spanning 10 CSS-related topics, to systematically evaluate the consistency of LLM-based simulations. Experiments reveal persistent role inconsistencies and show that better overall model performance does not imply higher simulation reliability. To improve consistency, this paper proposes AdaORPO, a reinforcement learning algorithm with an adaptive learning rate, which shows effectiveness across seven LLMs.

**Reasons To Accept:**

1.	The quality of the TRUSTSIM dataset is notably high—it has been curated through manual evaluation, LLM-based refinement, and a human panel review. This makes it a valuable contribution to the community.
2.	The experimental setup is robust. The authors evaluate 14 LLMs and provide detailed analyses, demonstrating a highly commendable level of experimental effort.
3.	In addition to introducing a high-quality dataset, the paper also proposes a solution, AdaORPO. The effectiveness of this method is well demonstrated through empirical results.

**Reasons To Reject:**

1. My only concern lies in the novelty of the dataset. It appears that the authors primarily expand the range of subjects to create a more extensive benchmark. While this is valuable, the dataset does not introduce a fundamentally new perspective or methodological innovation compared to prior work—its main distinction is the broader subject coverage.

---

> ### Author Response · Authors · 2025-05-29
> **Thanks for your comments**
>
> We sincerely thank the reviewer for their comments regarding the novelty of our dataset. We would like to clarify that our work goes far beyond merely expanding subject coverage, and we respectfully highlight the following key innovations:
>
> **a) Broader and More Complex Simulation Scenarios**:
> Unlike previous social simulation benchmarks, TRUSTSIM systematically covers a much wider range of topics—including Psychology, Sociology, Economics, Law, Ethics, Linguistics, etc.—and introduces significantly more complex and realistic scenarios. Our Table 1 demonstrates the diverse set of attributes (e.g., knowledge, capability, value, culture, time, social status) incorporated in each simulation instance, enabling a more granular and representative evaluation of LLMs in social character simulation.
>
> **b) Rigorous Data Quality Assurance**:
> The quality of TRUSTSIM is ensured through a multi-step process: human-AI collaborative scenario generation, manual instance composition by domain experts, LLM-powered refinement, and, most importantly, comprehensive human panel reviews. Each instance must be agreed upon by all four experts for semantic consistency, uniqueness, and representativeness, with only high-scoring items included (see Section 3.2–3.3). This level of quality control far exceeds prior datasets.
>
> **c) Extensive and Systematic Model Evaluation**:
> We conduct the largest-scale evaluation to date, testing 14 leading LLMs across all domains of TRUSTSIM. This allows us to draw important and generalizable conclusions about model performance, including identifying key trends, inconsistencies, and insights that were not accessible with previous benchmarks (see Section 5).
>
> **d) New Method for Reliability Enhancement**:
> Beyond dataset construction, we also propose AdaORPO—a novel adaptive learning rate algorithm for optimizing simulation consistency. This method is validated on multiple open-weight models, showing consistent improvements and offering a promising direction for future research (see Section 6).
>
> In summary, our work not only provides the most comprehensive and high-quality dataset for social character simulation to date, but also establishes a rigorous evaluation standard, delivers important empirical findings, and introduces a new alignment method, laying the groundwork for future advances. We hope this clarifies the multifaceted novelty and impact of our work.

---

> > ### Comment · Reviewer_Lamc · 2025-06-07
> >
> > Thanks for your reply. This is a solid paper; I will retain my score and recommend it for acceptance.

---

> > > ### Author Response · Authors · 2025-06-08
> > >
> > > Thank you very much for your positive feedback and recommendation. We appreciate your time and thoughtful review of our work!

---

### Official Review · Reviewer_2Gns · 2025-05-13

**Rating:** 6
**Confidence:** 4
**Ethics Flag:** 1

**Summary:**

This paper addresses an important issues of self-consistency and persona-consistency in LLMs. The paper introduces a new dataset of 740 instances from 10 categories relating to social sciences where each instance describes a particular scenario and persona and two questions for this combination. LLMs are then prompted with the question and then other LLMs are tasked with evaluating whether their responses are consistent with the prompt and internally with respect to the two questions' answers. The authors show that models are good at this benchmark, but there is room for improvement. To close the gap, the authors propose an RL method that aligns models for consistency, showing improvements in open-weight models.

The paper's topic is important but I found the execution hard to follow at times. Multiple works have already examined self consistency in LLMs and the paper does not go into detail about how this new work differs or adds to the discussion. For example, I would have liked to see some discussion in light of papers like Li et al. ICLR 2023 (https://arxiv.org/pdf/2310.01846) that use the same type of two consistency metrics.

The new dataset is potentially useful but the paper does not go into any detail about the implications for performance variation across domains, so it is not clear why the domains matter; in my reading, the domains could have all been collapsed together as an aggregate statistic and little would have changed in the paper's reading or implications.

**Questions To Authors:**

- In Table 1, I was not clear what these texts are, as "attributes" is not really references elsewhere in a way that shows how this text might be used in a prompt

- Line 143, did humans add much elaboration to the existing prompts?

- Line 146: does this mean that all characters are unique in the dataset?

- Lines 151: could you say more about how online news influenced the design? I appreciate the care for creating attributes but an example would really help the reader understand what the design criteria were.

- Line 159: I wasn't sure why ChatGPT-4o was needed to improve the readability scenarios. Why not keep them as-is in their natural form?

- Line 165: how many items were filtered out for being too similar?

- Line 188: Are there two annotation interfaces? It looks like one is a custom too but there's a second figure that looks like a google form so I wasn't sure how these were related (e.g., are both done at the same time?).

- Line 196: Are all three models used as judges? If so, how are labels aggregated?

- Line 206-210: It would help me, the reader, out a lot if either (i) the terms "general consistency" and "internal consistency" were used as terms throughout since these are bolded or (ii) the terms "satisfaction rate" and "inconsistency rate" replace the previous two terms as bolded text, since these are the two terms used in practice. It was very confusing to figure out where the metrics are defined without more searching.

- Table 2: I am not sure what rating score means with respect to your defined metrics.

**Reasons To Accept:**

- Important problem give the growing use of LLMs in social simulations

- Introduces a new dataset of scenarios and questions for testing model consistency

- Demonstrates a clear gap by current models and then introduces a new RL method to close that gap.

- Extensive testing of multiple models, both open and closed weight

**Reasons To Reject:**

- The dataset is relatively small and despite the effort to make the dataset diverse across social science fields, these differences are largely not discussed.

- Many methodological details are omitted and it was generally hard to follow the paper. Much detail is pushed to the appendix, which makes it very hard for readers. For example, when describing the RL training in 6.1, it is never described how the prompts are split in to train or test (I think this is described in line 600 in the appendix potentially).

- The paper's framing is concerned with simulation. However, the simulation here is very limited: answering just a single question. The related work is more concerned with multi-turn or multi-agent simulation. It would have been helpful to connect the current performance to these more general applications: do the scores on this new dataset predict downstream performance in these more complex settings?

- The authors have clearly done a lot of work, but at times, it's not really clear what the implications are for the paper. Some sections are mostly  descriptions of the contents of tables/figures (e.g., Lines 231-257) and I'm not sure what I should learn or change in my practice based on statements about how differently models perform. I do think the authors could address this, so this is not to say that the paper can't have deeper implications, just that they're not present to me in my reading.

---

> ### Author Response · Authors · 2025-05-29
> **Thanks for your comments**
>
> Thank you very much for your detailed feedback and thoughtful suggestions. We address your points as follows:
>
> - **Metrics Differences**: Li et al. ICLR 2023 measures whether a model's answer and its validation are logically consistent (GV consistency), while our paper focuses on whether a model's behavior aligns with its self-reported traits in social role simulations (role consistency).
>
> - **Domain Differences Not Discussed && Limited Dataset Size**:
> Thank you for the suggestion. While TRUSTSIM spans diverse domains with tailored scenarios (Section 3.1, Figure 3, Table 1) and reports performance by domain (Table 2, Figures 6–7), we agree that the impact of domain variation merits deeper discussion. In the revision, we will explicitly analyze domain-specific trends, highlight particularly challenging areas, and add examples to illustrate the role of domain context. We acknowledge that our dataset is relatively small due to the high construction cost—each instance involves detailed scenarios and complex human validation. We prioritized quality over quantity, which made large-scale creation more challenging.
>
> - **Methodological Clarity**:
> Thank you for noting the need for clearer methodological explanations. While details are in the main text and appendix (e.g., data splits in Appendix C), we agree that summarizing key points in the main text will help readers. We will make these clarifications and add cross-references in the revised draft.
>
> - **Extension to Multi-turn/Multi-agent Settings**:
> We agree this is an important direction. Although our current dataset focuses on single-turn scenarios, our design principles and methodology can be extended to multi-turn and multi-agent simulations. We will highlight this extensibility in the revision.
>
> - **Discussion of Impact**:
> Thank you for emphasizing the importance of actionable guidance and practical implications. Our findings are intended to inform both practitioners and researchers in several concrete ways:
>
>   - Model Selection: Differences in simulation consistency help users choose models better suited for social science tasks, beyond general benchmarks.
>
>   - Development Focus: Improving consistency requires more than larger models—targeted methods like AdaORPO offer significant gains and can guide more reliable LLM development.
>
>   - Benchmarking and Evaluation: Our dataset and approach offer a reusable framework for evaluating LLMs in role-based simulations across various domains.
>
> In response to your feedback, we have added a new "Discussion" section to the revised draft, where we summarize the practical impacts of our work.
>
> Responses to Specific Questions and Points:
>
> - Attributes and Data Diversity:
> The dataset attributes illustrate the diversity of our data, though they are not directly evaluated.
>
> - Human Elaboration and Uniqueness:
> Prompts were thoroughly elaborated by humans to ensure authenticity, and all roles are unique.
>
> - Providing Concrete Examples Without External Links:
> For instance, when designing a scenario in the domain of organizational behavior, we incorporated elements from some news reports about remote work trends and challenges faced by employees during the pandemic. This enabled us to create scenarios in which characters must navigate remote team management, reflecting both typical organizational concerns and current, real-world issues. We will add such concrete examples in the revised manuscript.
>
> - Use of GPT-4o:
> GPT-4o was employed solely to enhance the clarity and readability of scenario descriptions, without altering their substantive content. Expert-authored drafts sometimes contained complex or ambiguous language that could affect consistent interpretation by both models and human evaluators.
>
> - Data Filtering:
> According to our records, a total of 63 instances were filtered out for being overly similar.
>
> - Annotation Interfaces:
> Indeed, two different annotation interfaces were used, each corresponding to a specific quality control criterion in our human panel review process. These interfaces were not used simultaneously but rather sequentially.
>
> - LLM Judges Aggregation:
> All three LLMs were used as judges, with scores averaged or by majority vote; we will clarify this in the paper.
>
> - Metric Consistency:
> We will use metric names (“satisfaction rate” and “inconsistency rate”) consistently, and clarify the meaning of the “rating score” in Table 2.
>
> - Clarification of Table 2:
> Thank you for pointing this out. In Table 2, the "rating score" refers to the average score (on a 1–5 scale) assigned by the LLM judge(s) for open-ended responses, as part of our score-based evaluation metric (see Section 4). We have revised the table caption.
>
> Thank you again for your invaluable feedback, which has directly improved the clarity and usefulness of our work. Please feel free to reply if you'd like more details. Thanks!

---

> ### Author Response · Authors · 2025-06-11
>
> With the review period closing in less than 10 hours, we wanted to respectfully follow up in case you had any additional questions or feedback. We truly appreciate your thoughtful engagement with our submission and thank you again for your time. We look forward to hearing from you!

---

### Official Review · Reviewer_vTuT · 2025-05-13

**Rating:** 7
**Confidence:** 3
**Ethics Flag:** 1

**Summary:**

This paper contributes TrustSim, a benchmark to help the research community study the reliability of LLM-based social character simulations. The paper makes two types of contributions: (1) using TrustSim to identify inconsistencies in simulated roles and (2) proposing the AdaORPO algorithm to improve simulation consistencies.

**Questions To Authors:**

Clarifications on the TrustSim dataset:

Section 3.2 states that "human experts" took outlines from GPT-4o and developed more in-depth scenarios. Who were these human experts, from where were they sourced, and how was their expertise ascertained? Were they paid and, if so, how much?

The paper could benefit by clarifiying these aspects of the human role/process in annotation.

**Reasons To Accept:**

The TrustSim dataset, itself, appears to have been thoughtfully developed to encompass multiple key dimensions of social science research. The dataset comes with 740 instances across 10 dimensions.

The ablation analysis for AdaORPO seems fairly rigorous across 7 models, demonstrating the utility of Ada vs models without this approach. These results could be of interest to the broader community of people investigating adaptive learning techniques to improve model reliability and other applications.

**Reasons To Reject:**

The following are not necessarily reasons to "reject" but are ways in which the paper could be strengthened.

It would be useful for this paper to cite and contextualize some critical prior work in social character simulation in the related works section:

https://www.researchsquare.com/article/rs-3296728/v1

https://arxiv.org/abs/2310.11667

https://arxiv.org/abs/2312.03664

---

> ### Author Response · Authors · 2025-05-29
> **Thanks for your comments**
>
> Thank you very much for your valuable feedback. We address your concerns point by point as follows:
>
> - Missing References:
> We appreciate your suggestion regarding the missing references. After a careful review of the recommended papers, we agree that they are indeed relevant to our topic. We apologize for the previous oversight and have now updated our draft to include these references.
>
> - Details on Human Experts:
> As specified in lines 621–631 of the revised draft, the human experts involved in our study are PhDs in either sociology or computer science, representing a diversity of regions. We plan to disclose further details (such as affiliations) upon acceptance of the paper and with their consent. No compensation was provided to these experts, as they are also authors of this work.
>
> Thank you again for your insightful suggestions, which have contributed to a more comprehensive and transparent manuscript.

---

> > ### Comment · Reviewer_vTuT · 2025-06-10
> >
> > Thank you for your response! I will keep my score (accept), and I think this paper can be useful for the community of people studying the ability of LLMs to simulate human behavior.

---

> > > ### Author Response · Authors · 2025-06-10
> > >
> > > Thank you very much for your positive feedback and recommendation. We appreciate your time and thoughtful review of our work!

---

### Decision · Program_Chairs · 2025-07-08

**Decision:**

Accept

**Comment:**

This work presents a new dataset of 740 instances that was carefully created through manual evaluation, LLM-based refinement and unanimous agreement from a human panel. This likely resulted in a very high quality dataset. The work measures the social consistency of several closed and open source LLMs showing there is room for improvement on this benchmark. Finally, the work proposes a novel RL method to improve performance on the benchmark.

Pros:
- The work addresses an important problem given the increasing use of LLMs to simulate social situations.
- The presented dataset is very high quality and has gone through rigorous vetting.
- There is testing of both closed and open source models.

Cons:
- The paper is hard to follow with much of the details pushed to the appendix.
- The paper is framed as a simulation problem however the dataset instances only ask a single question rather than multi-turn/multi-agent scenarios in other work.